# Comparative evaluation of bioinformatic tools for virus-host prediction and their application to a highly diverse community in the Cuatro Ciénegas Basin, Mexico

**Alejandro Miguel Cisneros-Martínez**[1,2], **Ulises E. Rodriguez-Cruz**[1,2], **Luis D. Alcaraz**[3], **Arturo Becerra**[4], **Luis E. Eguiarte**[1], **Valeria Souza**[1,5]*

1 Departamento de Ecología Evolutiva, Instituto de Ecología, Universidad Nacional Autónoma de México, Ciudad de México, México, 2 Doctorado en Ciencias Biomédicas, Universidad Nacional Autónoma de México, Ciudad de México, México, 3 Departamento de Biología Celular, Facultad de Ciencias, Universidad Nacional Autónoma de México, Ciudad de México, México, 4 Departamento de Biología Evolutiva, Facultad de Ciencias, Universidad Nacional Autónoma de México, Ciudad de México, México, 5 Centro de Estudios del Cuaternario de Fuego-Patagonia y Antártica (CEQUA), Punta Arenas, Chile

* souza@unam.mx

**Data Availability Statement:** Supporting data are provided as supplementary data files. S1 File, S4

## Abstract

Due to the enormous diversity of non-culturable viruses, new viruses must be characterized using culture-independent techniques. The associated host is an important phenotypic feature that can be inferred from metagenomic viral contigs thanks to the development of several bioinformatic tools. Here, we compare the performance of recently developed virus-host prediction tools on a dataset of 1,046 virus-host pairs and then apply the best-performing tools to a metagenomic dataset derived from a highly diverse transiently hypersaline site known as the Archaean Domes (AD) within the Cuatro Ciénegas Basin, Coahuila, Mexico. Among host-dependent methods, alignment-based approaches had a precision of 66.07% and a sensitivity of 24.76%, while alignment-free methods had an average precision of 75.7% and a sensitivity of 57.5%. RaFAH, a virus-dependent alignment-based tool, had the best overall performance (F1_score = 95.7%). However, when predicting the host of AD viruses, methods based on public reference databases (such as RaFAH) showed lower inter-method agreement than host-dependent methods run against custom databases constructed from prokaryotes inhabiting AD. Methods based on custom databases also showed the greatest agreement between the source environment and the predicted host taxonomy, habitat, lifestyle, or metabolism. This highlights the value of including custom data when predicting hosts on a highly diverse metagenomic dataset, and suggests that using a combination of methods and qualitative validations related to the source environment and predicted host biology can increase the number of correct predictions. Finally, these predictions suggest that AD viruses infect halophilic archaea as well as a variety of bacteria that may be halophilic, halotolerant, alkaliphilic, thermophilic, oligotrophic, sulfate-reducing, or marine, which is consistent with the specific environment and the known geological and biological evolution of the Cuatro Ciénegas Basin and its microorganisms.

File, and S5 File are available with the online version of this article. The minimal data set required to replicate the reported study findings in their entirety, constituted by S2 File and S3 File, is available via the public Figshare repository: https://doi.org/10.6084/m9.figshare.24649935.v2. Metagenomic viral contigs and metagenome-assembled genomes have been submitted to GenBank under BioProject accession PRJNA847603.

**Funding:** AMCM is a Ph.D. student in the Programa de Doctorado en Ciencias Biomédicas, Universidad Nacional Autónoma de México (UNAM) and has received the Consejo Nacional de Humanidades, Ciencias y Tecnologías (CONAHCYT) grant 814975. UERC is a Ph.D. student in the Programa de Doctorado en Ciencias Biomédicas, Universidad Nacional Autónoma de México (UNAM) and has received the Consejo Nacional de Humanidades, Ciencias y Tecnologías (CONAHCYT) grant 857544. This study was supported by Programa de Apoyo a Proyectos de Investigación e Innovación Tecnológica (PAPIIT) – Dirección General de Asuntos del Personal Académico (DGAPA) - UNAM (grant numbers IG200319 and IN204822). This manuscript was written with Chile Agencia Nacional de Investigación y Desarrollo (ANID) – Centro de Estudios del Cuaternario de Fuego-Patagonia y Antártica (CEQUA) R-20F0009 to VS and LEE as part of a larger comparison with ocean waters in the southern tip of the American continent. The funders had no role in study design, data collection and analysis, decision to publish, or preparation of the manuscript.

**Competing interests:** I have read the journal's policy and the authors of this manuscript have the following competing interests: LDA is an Academic Editor for this journal. This does not alter our adherence to the PLOS ONE policy on sharing data and materials.

## Introduction

The Cuatro Ciénegas Basin (CCB) is a threatened oasis in the Chihuahuan Desert, Mexico. It is known for its oligotrophic waters, which nevertheless host a great biological diversity, including bacteria that are phylogenetically related to marine bacteria, despite having been isolated from any ocean for tens of millions of years [1]. The presence of endemic microorganisms adapted to a stoichiometry reminiscent of the Late Precambrian Supereon [2], whose closest relatives are marine bacteria from which they diverged an estimated 770–680 and 202–160 million years ago, respectively [3, 4], suggest that CCB diversity has evolved as a result of the long-term environmental stability of a deep aquifer that replicates the conditions of an ancient ocean [4, 5]. Thus, CCB aquatic systems are considered analogues of an ancient ocean and models for the study of ecological and evolutionary processes that occurred millions of years ago, while their arid, saline, gypsum-rich soils are considered analogues of Martian environments that may have supported life at some point in their geological history [6, 7].

Within the CCB, there is a unique hypersaline and alkaline pond known as the Archaean Domes (AD). Its name is linked to the presence of flexible microbial mats that swell during the rainy season, forming dome-shaped structures due to the release of anoxic gases reminiscent of the Archaean eon, such as methane and hydrogen sulphide [8, 9]. At this site, more than 6000 amplicon sequence variants (ASVs) were identified in 10 samples collected at a scale of 1.5 m [9]. This diversity includes a high abundance of halotolerant bacteria, as well as halophilic and methanogenic archaea, which are rare in the rest of the CCB [8, 9]. Finally, a highly diverse viral community has recently been described, in which haloarchaeaviruses appear to be an essential component, and which does not behave like those from other hypersaline or high pH sites in the face of environmental fluctuations [10]. However, viruses in AD have yet to be fully characterised, which is essential to dissect the virus-host relationships and interactions that may drive microbial and viral diversity in such a unique site.

Until 1990, the International Committee on Taxonomy of Viruses (ICTV) requested detailed information on biological properties to describe and classify new viruses. These biological properties were observed either *in vitro* (*e.g.*, replication cycle, virion structure and antigenic relationships) or in natural host interactions (*e.g.*, pathogenicity, epidemiology, and host range) [11]. However, the development of DNA sequencing techniques, bioinformatic tools, and methods to study molecular evolution, now allows metagenomic analyses that can detect a vast diversity of unknown, non-culturable viruses which cannot be described in the traditional way. For example, of 488,130 viral populations defined from the Global Ocean Viromes 2.0 (GOV 2.0) dataset, which consists of sequencing data from samples collected from around the world's oceans (including samples from the Tara Oceans Global Oceanographic Research Expedition, the Malaspina Expedition and the Tara Oceans Polar Circle Expedition), only 10% could be taxonomically assigned to a known viral family [12]. As a result, the scientific community has proposed incorporating metagenomic data into the ICTV taxonomy by using phenotypic traits and phylogenetic information inferred from assembled sequences and genomes [11].

Given that viruses are heavily dependent on cellular machinery to carry out their replication cycle, and that most viruses are named based on the host they interact with, among all the phenotipic traits that can be inferred from a genomic sequence, host range is a key factor in indentifying a virus in terms of its habitat, and the biological processes in which it might be playing a role. To gain insights into the host range of unculturable viruses, various bioinformatic methods for virus-host prediction have been developed over the past couple of years [13]. The publication of these tools is typically accompanied by validation tests with estimates of precision and sensitivity, as well as comparisons with other methods. However, most publications use different databases and sometimes use published values to compare the precision of different methods directly [14].

Predicting the host from a viral sequence is not an easy task, which has been evidenced by the lack of precision or sensitivity of the different approaches. Generally speaking, homology-based methods can achieve high precision but suffer from low sensitivity. In contrast, sequence composition methods have greater sensitivity but struggle to make correct predictions [13].

Here, we set out to assemble viruses from AD metagenomes and predict their hosts as a fundamental step in their characterization. However, given that information on the performance of virus-host prediction tools is sparse, and most comparisons include only a handful of tools, we first reviewed virus-host prediction methods and tools, and performed comparative evaluations of some of the most popular or recently developed tools to select the tools we would use to predict AD viruses hosts.

## Materials and methods

### Benchmarking of bioinformatics tools for virus-host prediction

Genomes were selected by downloading three lists (S1 File): i) NCBI complete viral genomes (https://www.ncbi.nlm.nih.gov/genomes/GenomesGroup.cgi) filtered by host 'bacteria'; ii) Virus-hostDB tabular report (https://www.genome.jp/ftp/db/virushostdb/) and; iii) RefSeq release 217 catalog (https://ftp.ncbi.nlm.nih.gov/refseq/release/release-catalog/), filtered by complete genomic molecule, not plasmid.

A link was established between the three tables so that if the viral genome accession had a match in the virus-hostDB, it was checked if the host taxa had a match in the RefSeq catalog (for a given virus with known host, check if the host has a complete genome). We found 102 hosts (bacteria species) with complete genomes. Of these, 16 bacteria were represented by more than one genome (up to 9). In such cases all genomes were downloaded. This resulted in 1,029 phage genomes and 133 bacteria genomes, which were downloaded from NCBI (S2 File). Given that 9 viruses were associated to more than one host (up to 5), and that there were 37 bacteria infected by more than one virus (up to 620), we ended up with 1,046 virus-host pairs. Following the same selection criteria we obtained 7 virus-pairs comprising 7 archaea virus genomes and 5 archaea genomes. The performance of the virus-host prediction tools was evaluated at the genus level.

A custom script was developed (https://github.com/AleCisMar/CrisprCustomDB/blob/main/benchmarking/compare_real-estimated.pl) to compare predicted virus-host pairs with confirmed pairs, providing true positives (TP), false positives (FP) and false negatives (FN) for each prediction tool. TP represents the correct identification of true pairs, while FP represents the incorrect identification of pairs (type I error). Finally, FN includes both FPs and viruses with unassigned hosts (NA), which represent a type II error or failure to identify true pairs. Precision, sensitivity, and F1_score were calculated as follows:

Precision (Positive Predictive Value):

$$PPV = \frac{TP}{TP + FP}$$

Sensitivity (True Positive Rate):

$$TPR = \frac{TP}{TP + FN}$$

F1_score:

$$F1_{score} = 2\left(\frac{PPV * TPR}{PPV + TPR}\right)$$

To run each program, we followed the instructions provided by the developers choosing the parameters for which they are reported to perform the best [14–20].

Since HostPhinder [19], CrisprOpenDB [15], VirHostMatcher-Net [20], and RaFAH [18] rely on extensive precompiled reference databases, only the 1,029 phage genomes were used as input. Since PHP [17] provides a reference database with 60,105 potential hosts two estimations were made, one using such reference database, and another using the custom database with 133 bacterial genomes. VirHostMatcher [14] and WIsH [16] were run on the custom database (1,029 phage genomes and 133 bacterial genomes). For VirHostMatcher, two estimations were made, both with a score $\leq$ 0.25. The first selects the most frequent host within the top 30 with the most similar profiles, and the second selects the most frequent host within the top 5 with the most similar profiles. For VirHostMatcher-Net, two estimations were also made. One without score restriction and the other limited to predictions with a score > 0.95.

## Virus-host prediction on assembled metagenomic reads

**Sample collection and sequencing.** Sampling was carried out at the Archaean Domes (AD) of the Rancho Pozas Azules (26˚49'41.9" N, 102˚01'23.6" W), belonging to Pronatura Noreste, in the Cuatro Ciénegas Basin (CCB), Coahuila, in the North of Mexico, under SEMARNAT scientific permit number SGPA/DGVS/03121/15. Twelve samples were taken between 2016 and 2020. For microbial mats, seven surface samples (M1 –M6 and D0) were collected using a sterile scalpel dissection (8 cm$^2$ / 40 cm$^3$) and transferred to 50 mL conical tubes. 30 cm plastic tubes were used as sediment samplers to collect two additional microbial mat samples at 30 and 50 cm depth (D30 and D50). Three samples were collected at the shallow ellipsoid orange pools or orange circles (OC) [8–10]: one superficial water sample (C0) on a 50 mL conical tube and two more at depths of 30 and 50 cm (C30 and C50). All samples were stored in liquid nitrogen until processing.

DNA was extracted according to [21] at the Laboratorio de Evolución Molecular y Experimental of the Instituto de Ecología, Universidad Nacional Autónoma de México, in Mexico City. Briefly, the extractions followed a column-based protocol with a Fast DNA Spin Kit for Soil (MP Biomedical) [22]. Total DNA was sent to CINVESTAV-LANGEBIO, Irapuato, México, for shotgun sequencing with Illumina Mi-Seq paired-end 2x300 technology.

All sequence reads are available on the National Centre for Biotechnology Information (NCBI) Sequence Reads Archive (SRA) under the BioProject accession: PRJNA847603.

**Read processing and assembly of metagenomic viral contigs.** The read quality was assessed with FastQC v0.11.9 [23]. Adapter removal and quality filtering were performed with Trimmomatic v0.39 [24] using a sliding window of 4 base pairs excluding reads with an average quality of less than 30 and less than 20 nucleotides. Clean reads were assembled with SPAdes 3.15.2 [25] using the—metaviral option. The viralVerify and viralComplete scripts (included in the SPAdes package) were used to verify that the assembled contigs correspond to viral genomes and to assess genome completeness, respectively. The circularity of the viral contigs was checked. When necessary, the position of sequences was adjusted prior to gene prediction and annotation with the help of custom scripts (available at https://github.com/AleCisMar/GenomicTools) that make use of BLAST [26], EMBOSS [27], Prodigal [28], and HMMER [29].

**Read processing, assembly, and taxonomic assignment of metagenome-assembled genomes.** The quality of the raw data was assessed with FastQC (v0.11.8) [23] and filtered with Trimmomatic (v0.39) [24]. The reads were then assembled using MetaSPAdes (v3.15.3) [30], and the contigs obtained in the assembly were used to perform read binning or

clustering, which was performed with MaxBin2 (v2.2.7) [31] and MetaBat2 (v2.12.1) [32]. Binning refiner (v1.4.2) software [33] was used to evaluate the percentage of contamination in the bins. The integrity of the metagenome-assembled genomes (MAGs) was assessed using CheckM (v1.1.3) [34] with the default settings. This resulted in 329 (35%) high quality MAGs with > 70% completeness and < 10% contamination, and 611 (65%) MAGs with varying contamination and integrity values.

For taxonomic assignment and placement of MAGs on the phylogenetic tree of life, we used the program GTDB-tk (v1.6.0) [35], which identifies 122 and 120 marker genes of archaea and bacteria, respectively, using HMMER [29]. Briefly, genomes are assigned to the domain with the most identified marker genes. Selected domain-specific markers are aligned with HMMER, concatenated into a single multiple sequence alignment, and trimmed with the ∼5000-column bacteria or archaea mask used by GTDB [35].

**Implementation of virus-host prediction tools on metagenomic data.** As CrisprOpenDB is the first tool to standardize CRISPR spacer-based phage host prediction [15], and has higher precision and sensitivity than previous CRISPR spacer-based host prediction pipelines, we wanted to use it to predict the hosts of metagenomic viral contigs (mVCs) from AD. However, we noticed that most of its predictions did not match the expected hosts given the environmental background of the sampling site. We thought that this might be related to the fact that no matter how extensive its spacer database is, if the actual hosts are uncharacterized bacteria or archaea, the database will not have the corresponding spacers for the predictions. Therefore, we thought that by predicting spacers from MAGs from AD and incorporating them into the large spacer database provided by Dion et al. [15], we would be able to make more and more accurate host predictions. However, CrisprOpenDB does not provide an easy way to add our own spacers to the database of over 11 million spacers, so we decided to develop a simple script that is inspired by CrisprOpenDB but allows to use custom databases in an easy way (CrisprCustomDB available at https://github.com/AleCisMar/CrisprCustomDB).

To run CrisprCustomDB, spacers must be predicted using the CRISPRDetect tool [36] (using an array_quality_cutoff of 3 –as recommended for FASTA files–). In Dion et al. [15] they also use the CRISPRDetect tool to predict spacers because it is one of the few tools that predicts the 5′-3′ orientation of the locus. CRISPRDetect generated 1,062 spacers (S3 File) from all 940 MAGs. CrisprCustomDB consists of two scripts: get_blast_tables.sh and get_host_id.pl. The get_blast_tables.sh script reads the.gff file generated by CRISPRDetect to extract the spacers in FASTA format (1,039 spacers with sequence lengths ranging from 28 to 43 bp). A BLAST nucleotide database is then created and the mVCs (all sequences in a single multi-FASTA file) are searched with Blastn. From the tabular output of Blastn, it keeps matches without gaps and with a maximum of 2 nucleotide mismatches (true mismatches = spacer length—alignment length + Blastn mismatches). The get_host_id.pl script reads the Blastn output tabular format created in the previous step. First, it determines whether a query mVC matches spacers from more than one possible host. If it has only one match or multiple matches to a single host, that host is assigned (criterion 1). If it has multiple matches to more than one possible host, it estimates for each possible host the number of spacers that match non-overlapping regions in the mVC. The host with more spacers matching non-overlapping regions is assigned (criterion 2). If possible hosts have the same number of spacers matching non-overlapping regions, it calculates the spacer start position relative to the corresponding CRISPR array start position:

$$relative\ position = \frac{spacer\ start - array\ start}{array\ end - array\ start}$$

So that it ranges from 0 (closer to the 5′ end) to 1. The host with the spacer closest to the 5′ end is assigned (criterion 3).

To illustrate the benefit of using the above filtering and host assignment criteria, we performed an additional CRISPR spacer-based host prediction using another CRISPR spacer prediction tool. For this prediction, spacers were detected using the CRISPRCasTyper program (v 1.3.0) [37] with the following parameters: cctyper -t 4—prodigal single—circular. This tool was more sensitive than the CRISPRDetect tool and produced 2,660 spacers (S3 File). All spacers were retained regardless of their length. The mVCs were run against this spacer database using Blastn, allowing a maximum of 2 mismatches (as calculated by Blastn). No other criteria were used to assign possible hosts.

Virus-host predictions were also made using CrisprOpenDB [15], RaFAH [18] and PHP [17]. As CrisprOpenDB and RaFAH provide extensive pre-computed databases, they only take mVCs as input. For PHP, k-mer frequencies were calculated for all 940 MAGs (S3 File).

## Results

### Classification of virus-host prediction methods

From the literature we could glimpse a five-category classification [38] (Fig 1): i) host-dependent alignment-based methods; ii) host-dependent alignment-free methods; iii) virus-dependent alignment-based methods; iv) virus-dependent alignment-free methods; and v) integrative methods. Host-dependent alignment-based methods include methods based on homology signals, such as searching for homology between viral and host proteins, tRNAs, viral genomes and CRISPR spacers, integrated prophages, and protein-protein interactions (PPI). These methods are helpful for detecting recent infections but have the disadvantage that not all viruses share genes with their hosts, which tends to make them precise, but with a low detection rate [13]. CrisprOpenDB is a recently released tool that uses biological criteria to standardize host predictions based on CRISPR spacers with increased sensitivity and precision thanks to its >11 million spacers database derived from >300,000 candidate hosts [15].

Host-dependent alignment-free methods include those based on sequence composition (e.g., similarity in codon usage, similarity in oligonucleotide composition, and GC content), which rely on the notion that viruses, being genetic parasites, approximate their nucleotide composition to that of the host over time. This genomic mimicry may allow viruses to use the

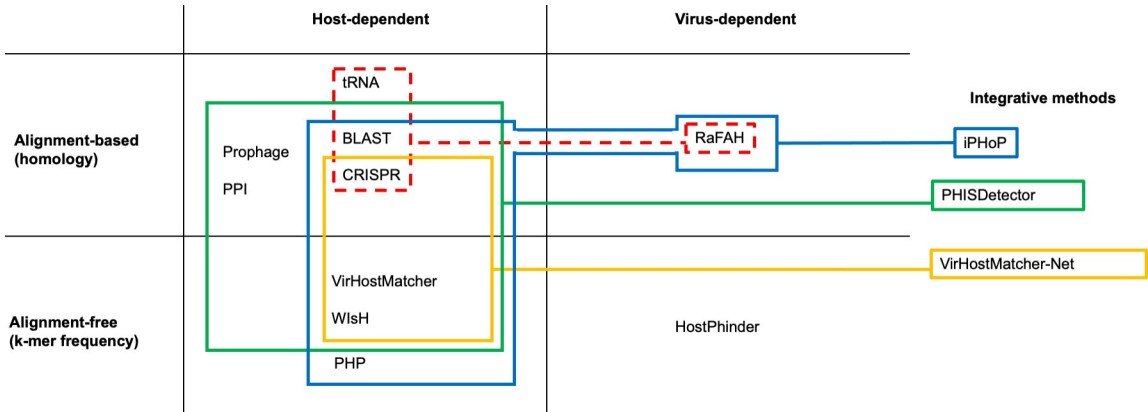

**Fig 1. Classification of virus-host prediction methods.** RaFAH uses host-dependent alignment-based methods to build part of its training database (red discontinuous lines). Integrative methods (iPHoP–blue lines; PHISDetector–green lines; VirHostMatcher-Net–yellow lines) attempt to exploit the virtues of a different number of methods. PPI = protein-protein interactions.

same tRNAs for protein synthesis or to evade the detection and degradation mechanisms of foreign nucleic acids. However, viruses can have similar sequence profiles independently, which can lead to a high false positive rate [13]. VirHostMatcher, which evaluates virus-host genome similarity through d*2 distance from 6-mer profiles [14], WIsH, which uses 8-mer profiles and Hidden Markov models (HMMs) [16] and PHP, which uses 4-mer profiles and a Gaussian model [17], are some well-known similar host-dependent alignment-free methods. These alignment-free strategies also include methods based on co-abundance profiles, which rely on the notion that viruses can only be found in the environment in which their host is also found. This profiling method requires the calculation of correlations of normalized abundance profiles of phage and bacteria in different environmental samples. However, they entail a major drawback: predator-prey interactions–such as those described by the kill-the-winner model [39, 40]–can generate positive or negative correlations, depending on where the interaction was at the time the sample was taken.

Instead of relying on host databases, virus-dependent methods depend on databases storing viruses with known hosts, to which query viruses are related either through homology signals (alignment-based) or their similarity in oligonucleotide composition (alignment-free). On the one hand, a machine-learning approach named Random Forest Assignment of Hosts (RaFAH) [18] is a virus-dependent alignment-based method that builds a part of its training database from CRISPR spacers, the presence of horizontally transferred genes and common tRNAs to ultimately associate the query virus to a virus with a known host through similarity in protein content. On the other hand, HostPhinder [19] is a virus-dependent alignment-free method that compares 16-mer profiles between query viruses and a database of 2,196 phages with known hosts.

Finally, integrative methods attempt to exploit the virtues of different methods like VirHostMatcher-Net [20], which integrates host-dependent alignment-based methods (CRISPR spacers) and host-dependent alignment-free methods (VirHostMatcher or WIsH) in a network framework, PHISDetector [41], which integrates BLAST [26], CRISPR spacers, prophage, and PPI analyses through a set of machine learning approaches, or iPHoP, which uses machine learning algorithms to compute taxonomy-aware scores for BLAST, CRISPR, VirHostMatcher, WIsH, and PHP, and integrates them with RaFAH results to obtain a final composite score [38].

## Benchmarking of bioinformatics tools for virus-host prediction

The best three performing tools for complete bacteria and phage genomes datasets (F1_score) were RaFAH, PHP, and VirHostMatcher-Net. They were followed by WIsH, VirHostMatcher, CrisprOpenDB, and HostPhinder at the bottom (Table 1).

CrisprOpenDB made 392 predictions, of which 259 were correctly estimated. These results translate into a sensitivity of 24.76%, a precision of 66.07%, and an F1_score of 36.02% (Fig 2).

Alignment-free methods evaluated here make predictions by comparing the oligonucleotide profile of a virus to either the oligonucleotide profile of viruses with a known host (HostPhinder) or the oligonucleotide profile of bacteria (VirHostMatcher, WIsH, PHP). Although HostPhinder predicted 1,044 pairs, most predictions were incorrect (677). Hence, it had the lowest performance of the alignment-free methods (Fig 2), with a sensitivity of 35.09%, a precision of 35.15%, and an F1_score of 35.12%.

VirHostMatcher was executed with two different criteria: i) selecting the most frequent host among the top thirty and; ii) selecting the most frequent host among the top five. When using the first criterion, VirHostMatcher generated more predictions (743 compared to 638) and produced more false positives (284 compared to 34). As a result, it achieved lower

**Table 1. Precision, sensitivity, and F1_score estimates of the different virus-host prediction tools.**

| Software | Actual virus-host pairs | Predicted pairs | NA | True positive | False positive | False negative | Precision | Sensitivity | F1_score |
|---|---|---|---|---|---|---|---|---|---|
| HostPhinder | 1046 | 1044 | 2 | 367 | 677 | 679 | 0.3515 | 0.3509 | 0.3512 |
| CrisprOpenDB | 1046 | 392 | 654 | 259 | 133 | 787 | 0.6607 | 0.2476 | 0.3602 |
| VirHostMatcher[†] | 1046 | 743 | 303 | 459 | 284 | 587 | 0.6178 | 0.4388 | 0.5131 |
| PHP[§] | 1046 | 1001 | 45 | 550 | 451 | 496 | 0.5495 | 0.5258 | 0.5374 |
| VirHostMatcher[¶] | 1046 | 638 | 408 | 604 | 34 | 442 | 0.9467 | 0.5774 | 0.7173 |
| WisH | 1046 | 1046 | 0 | 794 | 252 | 252 | 0.7591 | 0.7591 | 0.7591 |
| VirHostMatcher-Net[*] | 1046 | 903 | 143 | 829 | 74 | 217 | 0.9181 | 0.7925 | 0.8507 |
| VirHostMatcher-Net | 1046 | 1046 | 0 | 921 | 125 | 125 | 0.8805 | 0.8805 | 0.8805 |
| PHP | 1046 | 1046 | 0 | 952 | 94 | 94 | 0.9101 | 0.9101 | 0.9101 |
| RaFAH | 1046 | 1046 | 0 | 1001 | 45 | 45 | 0.957 | 0.957 | 0.957 |

[†]Prediction using score $\leq 0.25$ and selecting the most frequent host among top 30.

[§]Using PHP reference database with 60,105 prokaryotic genomes.

[¶]Prediction using score $\leq 0.25$ and selecting the most frequent host among top 5.

[*]Prediction using score $> 0.95$.

sensitivity (43.88% compared to 57.74%), precision (61.78% compared to 94.67%), and F1_score (51.31% compared to 71.73%).

Among these methods, WIsH and PHP emerged as the top predictors, achieving the maximum number of pairs (1,046). WIsH demonstrated a sensitivity, precision, and F1_score of 75.91%, whereas PHP appeared as the best-performing alignment-free method (Fig 2) with a sensitivity, precision, and F1_score of 91.01%. PHP was also tested against a reference database with 60,105 potential hosts provided by the authors. However, this test resulted in fewer predictions (1,001) and lower sensitivity (52.58%), precision (54.95%), and F1_score (53.74%).

VirHostMatcher-Net was executed using two approaches: first, by setting a prediction threshold with a score $> 0.95$ and, second, without any score restrictions. Restricting the final host assignment to predictions with higher scores resulted in higher accuracy (91.81% vs. 88.05%) at the expense of lower sensitivity (79.25% vs. 88.05%) and, as a consequence, a lower F1_score (85.07% vs. 88.05%). Meanwhile, RaFAH achieved an accuracy, sensitivity, and F1_score of 95.70%, making it the algorithm with the best overall performance (Fig 2).

## Virus-host predictions on metagenomic viral contigs from Archaean Domes, Cuatro Ciénegas Basin, Mexico

To predict the host of metagenomic viral contigs (mVCs) from Archaean Domes (AD) at Cuatro Ciénegas Basin (CCB), based on CRISPR spacers predicted on metagenome-assembled genomes (MAGs) from the same dataset, we employed two related approaches. The first approach involved conducting a Blastn search using 2,660 spacers, with a maximum of 2 mismatches as the only criterion. The second approach involved CrisprCustomDB, using 1,062 spacers to solve problematic host assignments. Additionally, we performed predictions using CrisprOpenDB, PHP, and RaFAH, as these tools demonstrated superior performance in their respective categories. Since HostPhinder showed lower performance than all other alignment-free methods, and PHP and RaFAH outperformed VirHostMatcher-Net, we did not make predictions with these tools. While the ordinary CRISPR approach, CrisprCustomDB, and PHP were executed on data derived from the AD MAGs, CrisprOpenDB and RaFAH only required the mVCs, as they relied on their extensive pre-compiled reference databases.

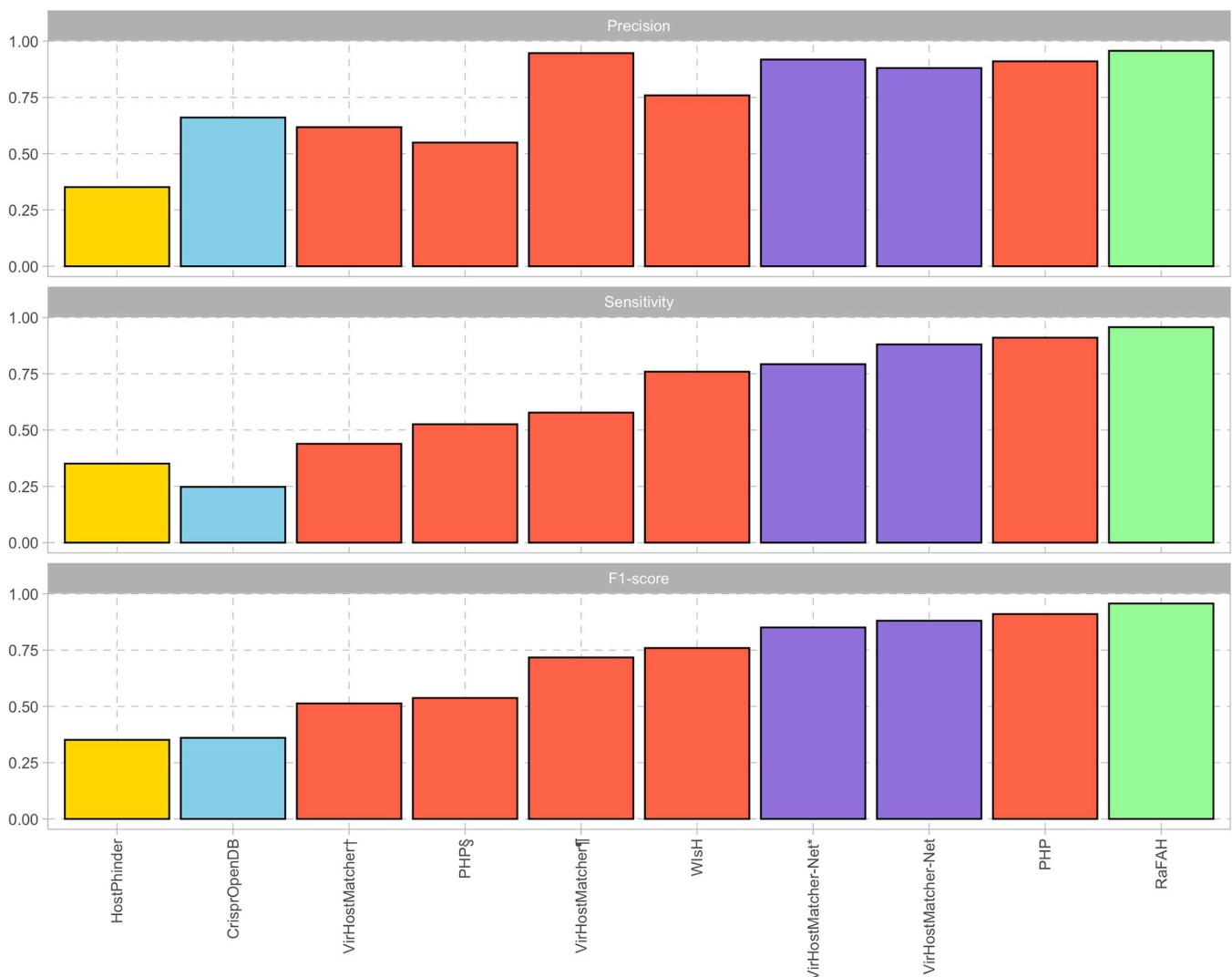

**Fig 2. Precision, sensitivity, and F1_score estimates of the different virus-host prediction tools.** VirHostMatcher† was tested with a score ≤ 0.25 and selected the most frequent host within the top 30. VirHostMatcher¶ was tested with the same parameters but selecting the most frequent host within the top 5. PHP§ was tested against a reference database of 60,105 potential hosts. For VirHostMatcher-Net*, only predictions with a score > 0.95 were kept. Bars are color-filled based on the method classification. Host-dependent, alignment-based = light blue; host-dependent, alignment-free = light red; virus-dependent, alignment-based = light green; virus-dependent, alignment-free = light orange; integrative = light purple.

Despite using spacer databases of radically different sizes, both CrisprCustomDB (1,062 spacers) and CrisprOpenDB (11,674,395 spacers) made only five predictions. The standard CRISPR approach made 8 predictions, while PHP and RaFAH made 54 and 87 predictions respectively (Fig 3). RaFAH had the lowest consistency with other methods. There were 84 predictions made by RaFAH alone (97.7% of its predictions), 45 by PHP alone (83.33% of its predictions) and 4 by CrisprOpenDB alone (80% of its predictions). All predictions made by CrisprCustomDB and the standard CRISPR approach were shared with other methods. Two of the predictions made by CrisprCustomDB were only consistent with the standard CRISPR approach, while the remaining three were consistent with both the standard CRISPR approach and PHP. CrisprOpenDB had only one prediction consistent with PHP. The standard CRISPR approach had two additional predictions consistent with PHP only and one consistent with

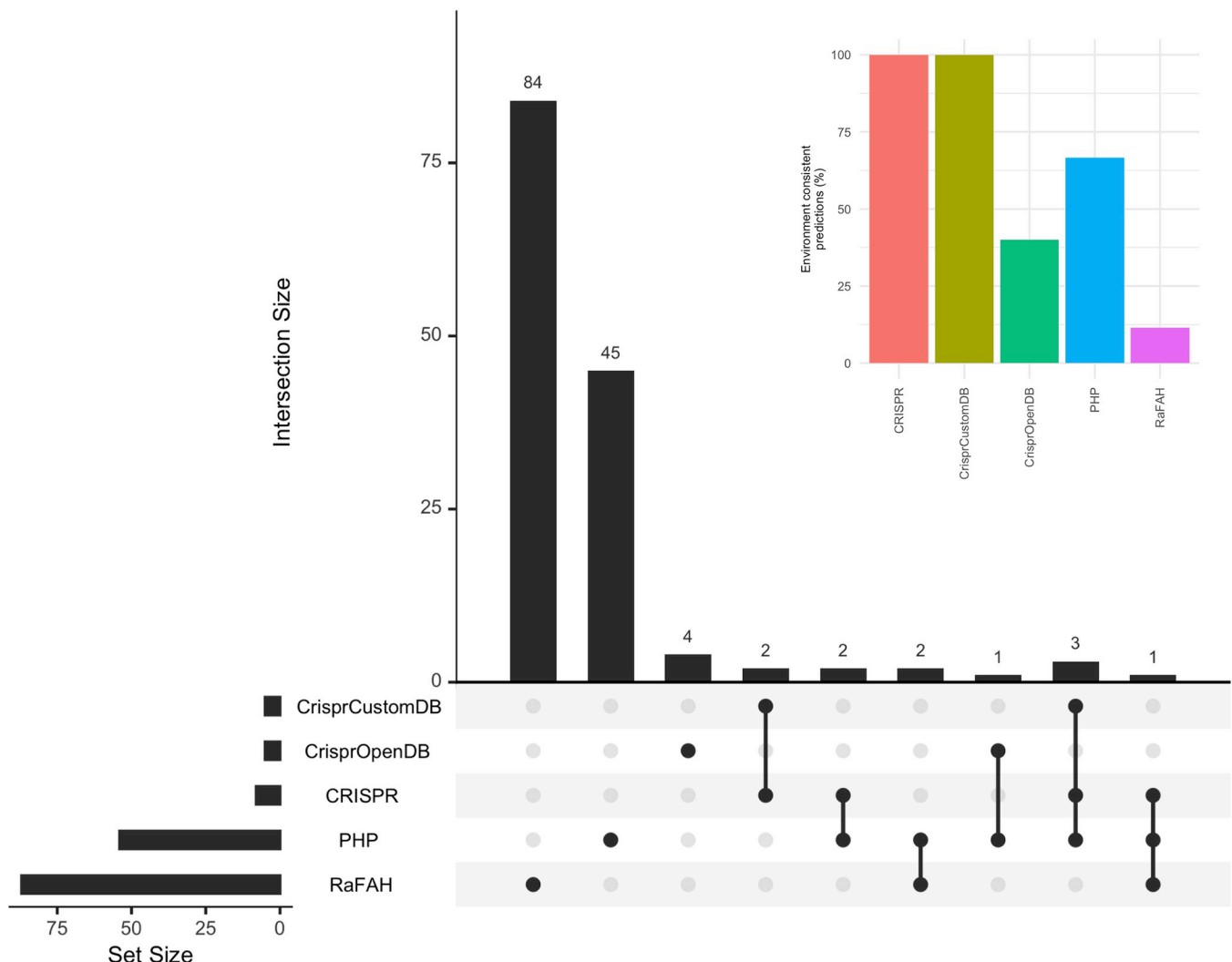

**Fig 3. Virus-host predictions consistent between methods or with the environment.** The main plot shows the number of consistent predictions between methods. The inset plot shows the percentage of predictions consistent with the environment. Set sizes: CrispCustomDB = 5; CrisprOpenDB = 5; CRISPR = 8; PHP = 54; RaFAH = 87.

both PHP and RaFAH. Finally, there were two additional predictions made by PHP that were shared with RaFAH (Fig 3). Although PHP had a low proportion of predictions consistent with other methods (16.67%), it had the highest number of shared predictions (9 predictions), followed by the standard CRISPR approach (8 predictions), CrisprCustomDB (5 predictions), RaFAH (3 predictions) and CrisprOpenDB (1 prediction).

For both standard CRISPR (8 predictions) and CrisprCustomDB (5 predictions), all predictions were consistent with what would be expected given known environmental conditions and types of organisms specific to AD (Fig 3). These included halophilic, sulfate-reducing, and sulfur-oxidizing bacteria (Table 2). CrisprOpenDB had only two predictions consistent with the environment (40% of its predictions) (halophilic and thiosulfate-reducing bacteria), while PHP had 36 (66.67% of its predictions, including several types of archaea, as well as halophilic, sulfate-reducing, sulfur-oxidizing, thermophilic, haloalkaliphilic, halotolerant, and marine bacteria) and RaFAH 10 (11. 49% of its predictions, including halophilic archaea, as well as

**Table 2. 46 host predictions on mVCs from Archaean Domes Pond, Cuatro Ciénegas, Mexico, designated as reliable according to different criteria.**

| Contig | CRISPR | CrisprCustomDB | CrisprOpenDB | PHP | RaFAH | Supporting evidence | Reference |
|---|---|---|---|---|---|---|---|
| C50N1L42 | *Desulfovibrionales; Desulfovermiculus* | NA | NA | *Desulfovibrionales; Desulfohalobiaceae* | *Desulfovibrionales; Desulfovibrio* | Halophilic; sulfate-reducing | [8, 9, 42] |
| M5N2L438 | *Halorhodospira* | *Halorhodospira* | NA | *Halorhodospira* | *Pseudomonas* | Halophilic | [43] |
| M6N1L439 | *Halorhodospira* | *Halorhodospira* | NA | *Halorhodospira* | *Pseudomonas* | Halophilic | [43] |
| C30N1L64 | *Thiohalorhabdus/ Thiohalospira* | *Thiohalorhabdus** | NA | *Thiohalorhabdus* | *Vibrio* | Halophilic; sulfur-oxidizing | [42, 44] |
| C0N5L506 | *Desulfobacterales* | NA | NA | *Desulfobacterales* | *Clostridium* | Sulfate-reducing | [8] |
| M1N5L607 | *Desulfobacterales* | *Desulfobacterales* | NA | NA | *Pseudoalteromonas* | Sulfate-reducing | [8] |
| C0N1L394 | *Desulfohalobiaceae; Desulfovermiculus* | NA | NA | *Desulfohalobiaceae* | *Bacteroides* | Halophilic; sulfate-reducing | [8, 9, 42] |
| M4N1L642 | *Halochromatium* | *Halochromatium* | *Thiobacillus* | NA | *Kingella* | Halophilic | [45] |
| C0N2L458 | NA | NA | *Gammaproteobacteria; Halomonas* | *Gammaproteobacteria; Halochromatium* | *Thauera* | Halophilic | [42, 45] |
| M5N6L415 | NA | NA | NA | *Archaea;Hadarchaeia* | *Archaea;Haloarcula* | Archaea | [8] |
| M6N2L524 | NA | NA | NA | *Halobacteriales; Halorubrum* | *Halobacteriales; Haloarcula* | Halophilic archaea | [8] |
| M1N1L790 | NA | NA | *Halanaerobium* | NA | *Clostridium* | Halophilic; thiosulfate-reducing | [8, 9, 42] |
| D30N111L | NA | NA | NA | *Archaeoglobaceae* | *Pseudomonas* | Archaea | [8] |
| D30N2L48 | NA | NA | NA | *Bathyarchaeia* | *Veillonella* | Archaea | [8] |
| D30N115L | NA | NA | NA | *Nanoarchaeia* | *Bacillus* | Archaea | [8] |
| M4N1L424 | NA | NA | NA | *Dichotomicrobium* | *Parabacteroides* | Thermohalophilic | [46] |
| D30N1L56 | NA | NA | NA | *Aminicenantaceae* | *Clostridium* | Deep marine sediments | [47] |
| M3N8L364 | NA | NA | NA | *Anaerolineae* | *Vibrio* | Deep marine sediments | [48] |
| M1N5L608 | NA | NA | NA | *Anaerolineae* | *Fusobacterium* | Deep marine sediments | [48] |
| M5N3L645 | NA | NA | NA | *Anaerolineae* | *Kingella* | Deep marine sediments | [48] |
| C50N2L80 | NA | NA | NA | *Anaerolineae* | *Vibrio* | Deep marine sediments | [48] |
| D50N2L80 | NA | NA | NA | *Anaerolineae* | *Vibrio* | Deep marine sediments | [48] |
| M5N8L404 | NA | NA | NA | *Bipolaricaulia* | *Leptotrichia* | Hypersaline sediments | [49] |
| M6N4L404 | NA | NA | NA | *Bipolaricaulia* | *Leptotrichia* | Hypersaline sediments | [49] |
| D30N50L3 | NA | NA | NA | *Bipolaricaulia* | *Vibrio* | Hypersaline sediments | [49] |
| C50N1L90 | NA | NA | NA | *Chitinivibrionales* | *Prevotella* | Haloalkaliphilic | [50] |
| M1N25L46 | NA | NA | NA | *Chitinivibrionales* | *Chlamydia* | Haloalkaliphilic | [50] |
| D30N6L39 | NA | NA | NA | *Chitinivibrionales* | *Porphyrobacter* | Haloalkaliphilic | [50] |
| M1N22L26 | NA | NA | NA | *Chitinivibrionales* | *Pseudomonas* | Haloalkaliphilic | [50] |
| M5N4L592 | NA | NA | NA | *Halothiobacillaceae* | *Faecalibacterium* | Halotolerant; halophilic | [51] |
| M6N2L592 | NA | NA | NA | *Halothiobacillaceae* | *Faecalibacterium* | Halotolerant; halophilic | [51] |
| M5N7L416 | NA | NA | NA | *Wenzhouxiangella* | *Burkholderia* | Haloalkaliphilic | [52] |
| M6N3L417 | NA | NA | NA | *Wenzhouxiangella* | *Burkholderia* | Haloalkaliphilic | [52] |

(*Continued*)

**Table 2.** (Continued)

| Contig | CRISPR | CrisprCustomDB | CrisprOpenDB | PHP | RaFAH | Supporting evidence | Reference |
|---|---|---|---|---|---|---|---|
| M1N1L521 | NA | NA | NA | *Halofilum* | *Vibrio* | Marine solar saltern | [53] |
| M5N28L50 | NA | NA | NA | *Gemmatimonadetes* | *Haloarcula* | Halophilic archaea | [8] |
| M6N4L511 | NA | NA | NA | *Gemmatimonadetes* | *Haloarcula* | Halophilic archaea | [8] |
| C0N2L195 | NA | NA | NA | *Halanaerobiales* | *Alistipes* | Halophilic; thiosulfate-reducing | [8, 9, 42] |
| M4N3L527 | NA | NA | NA | *Phycisphaerales* | *Pseudomonas* | Marine | [54] |
| M1N3L461 | NA | NA | NA | *Rhodothermales* | *Alistipes* | Thermohalophilic; haloalkaliphilic | [55, 56] |
| D30N26L5 | NA | NA | NA | *Petrotogales* | *Fusobacterium* | Thermophilic | [57] |
| C0N1L567 | NA | NA | NA | *Halanaerobiales* | *Clostridium* | Halophilic | [58] |
| C30N1L45 | NA | NA | NA | NA | *Salinispora* | Marine sediments | [59] |
| M1N6L535 | NA | NA | NA | NA | *Desulfotomaculum* | Thermophilic; sulfate-reducing | [60] |
| M1N8L483 | NA | NA | NA | NA | *Thermus* | Thermophilic | [61] |
| M5N19L71 | NA | NA | NA | NA | *Caulobacter* | Oligotrophic | [62] |
| M6N6L714 | NA | NA | NA | NA | *Caulobacter* | Oligotrophic | [62] |

Reliable predictions, either through consistency between methods or consistency between the source environment and the predicted host biology (taxonomy, lifestyle, or metabolism), are underlined. Where applicable, the lowest common taxonomic rank and the lowest taxonomic rank achieved by each tool are separated by ";". The full list of predictions can be found in the S4 File.

*Assigned using criterion 3: Multiple hosts matching the same number of regions. Host with spacer closest to the 5' end.

halophilic, sulfate-reducing, thermophilic, oligotrophic, and marine bacteria) predictions consistent with the environment. PHP also had the highest number of predictions consistent with the environment, followed by RaFAH, the standard CRISPR approach, CrisprCustomDB, and CrisprOpenDB.

Consistency between the source environment and the predicted host biology (taxonomy, habitat, lifestyle, or metabolism) may indicate a true prediction. However, the likelihood of a true prediction increases with support from a greater number of tools. Of the 46 predictions consistent with the environment, there were 35 predictions supported by only one method, 7 predictions supported by two methods, and 4 predictions supported by three methods (Table 2). The latter include contig C50N1L42 assigned to *Desulfobacterota* by ordinary CRISPR, PHP, and RaFAH, and three contigs assigned to *Proteobacteria* (contigs M5N2L438 and M6N1L439 assigned to *Halorhodospira*, and contig C30N1L64 assigned to *Thiohalorhabdus*) by ordinary CRISPR, CrisprCustomDB, and PHP. It is worth noting that the ordinary CRISPR approach assigned two candidate hosts for contig C30N1L64 (*Thiohalorhabdus* / *Thiohalospira*), which was successfully solved by CrisprCustomDB by assigning *Thiohalorhabdus* as the candidate host with the spacer closest to the 5′ end. Such a prediction was also supported by PHP. Predictions supported by two methods include contigs C0N5L506 and C0N1L394 assigned to *Desulfobacterales* and *Desulfohalobiaceae*, respectively, by CRISPR ordinary and PHP, contigs M1N5L607 and M4N1L642 assigned to *Desulfobacterales* and *Halochromatium*, respectively, by CRISPR ordinary and CrisprCustomDB, contig C0N2L458 assigned to *Gammaproteobacteria* by CrisprOpenDB and PHP, and contigs M5N6L415 and M6N2L524 assigned to *Archaea* and *Halobacteriales* by PHP and RaFAH, respectively.

If we consider predictions consistent with the environment to be true positives, RaFAH would be the tool with the lowest precision (11.5%), followed by CrisprOpenDB (40%), PHP (66.67%), and ordinary CRISPR and CrisprCustomDB (both with 100%). In this respect, the

ordinary CRISPR approach, CrisprCustomDB, and PHP, all of which were run against custom databases derived from MAGs, seem to have better performance compared to RaFAH and CrisprOpenDB, which use extensive pre-compiled databases of viruses with known host and known spacers, respectively. However, all CRISPR spacer-based tools seem to lack sensitivity compared to RaFAH and PHP. Finally, it is worth mentioning that RaFAH was the only tool that correctly predicted the host of Escherichia virus ΦX174, which was used as a positive control for DNA sequencing (S4 File).

## Discussion

The increasing number of virus-host prediction tools prompted us to perform a comparative evaluation of the most popular and recently released tools (Fig 1). Optimization of precision and sensitivity estimates within each approach has been achieved by either using more extensive reference databases (*e.g.*, CrisprOpenDB), by leveraging the power of different machine learning algorithms (PHP, RaFAH, VirHostMatcher-Net, PHISDetector, iPHoP), or by integrating different methods (VirHostMatcher-Net, PHISDetector, iPHoP). The publication of these tools is typically accompanied by validation tests with estimates of precision and sensitivity, as well as comparisons with other methods. However, most publications use different databases and sometimes use published values to compare the precision of different methods [14] directly. So far, Roux et al. [38] have compared the largest number of methods showing that host-dependent alignment-based methods can achieve high precision but suffer from low sensitivity. In contrast, host-dependent alignment-free methods have greater sensitivity but struggle to make correct predictions, while virus-dependent alignment-based methods such as RaFAH present both high sensitivity and precision. However, virus-dependent methods may underperform when predicting the host of novel viruses, which also affects, to a lesser extent, host-dependent alignment-based methods but not alignment-free methods [38].

Unfortunately, due to disk space limitations and to the size of the databases, we could not evaluate either PHISDetector [41] or iPHoP [38]. Since there is a discrepancy in the performance of PHISDetector compared to VirHostMatcher-Net [38, 41], we can only conclude which tool performs the best once we compare them under the same methodological framework. As for iPHoP, this is probably the best-performing integrative tool [38], as it integrates RaFAH into its host prediction algorithm, which has shown better performance than VirHostMatcher-Net both here (Fig 2) and in its original publication [18].

According to the literature, it is understood that following iPHoP, PHISDetector (compared with VirHostMatcher-Net, PHP, WIsH and VirHostMatcher) [41] and RaFAH (reported with higher F1_score than the combination of CRISPR, BLAST and tRNAs, followed by VirHostMatcher-Net, WIsH, HostPhinder, and CRISPR, BLAST and tRNAs individually) [18] are the most precise tools. They are likely to be followed by VirHostMatcher-Net (more precise than similarity networks, CRISPR, BLAST, WIsH, and VirHostMatcher) [20], PHP (reported less precise than CRISPR and BLAST, which, however, have very low sensitivity, but are more precise than WIsH and VirHostMatcher) [17] and WIsH (reported to be more precise than VirHostMatcher, especially for incomplete or short viral genomes) [16]. Lastly, CrisprOpenDB (reported to have similar precision to WIsH) [15], HostPhinder (reported to be more precise than BLAST) [19], and VirHostMatcher (compared to values published by Edwards et al. [13] appears to have similar precision to homology methods—BLAST, prophage, and CRISPR—and higher than early implementations of the k-mer method, abundance profiling and GC content) [14] appear to be the least precise tools.

To test the above interpretations about the performance of virus-host prediction tools, we downloaded 1,029 and 133 complete phages and bacterial genomes, respectively. (S1 and

S2 Files), making up 1,046 virus-host pairs. We did not present the results of archaea viruses and their respective hosts, because we could only retrieve seven pairs following the method described in the Materials and Methods section (see results in S5 File where host-dependent, alignment-free methods show the highest precision, sensitivity and F1_scores). In addition, some of the virus-host prediction tools evaluated here are explicitly trained on bacteria and their corresponding phages (*e.g.* [15]) and, therefore, cannot be used to evaluate their performance on viruses of archaea. The performance of the virus-host prediction tools was evaluated at the genus level because performance comparisons are often consistent across taxonomic ranks [14, 16–18, 20, 41], and because it may be more biologically informative than higher taxonomic rank predictions.

As expected [13, 17, 38], CrisprOpenDB showed high precision at the expense of sensitivity. Although it had the lowest sensitivity of the tools compared, it had a significant increase in sensitivity compared to CrisprCustomDB (not shown). For CrisprCustomDB, 1,349 spacers (S2 File) were found in 40 of the 133 bacterial genomes (30%), but it only managed to make 28 predictions (sensitivity < 3%), which is 364 less than CrisprOpenDB. This demonstrates the benefit of using a database of > 11 million spacers. Although CrisprOpenDB has increased the sensitivity of CRISPR-based methods, they still need to catch up with newer sequence composition methods that have high sensitivity and improved precision. This was the case for VirHostMatcher, WIsH and PHP, which achieved sensitivity and precision >50%. In contrast, HostPhinder had a higher sensitivity than CRISPR-based methods, but the lowest precision of all methods compared. This result suggests that relying solely on transferring the host of the most similar virus may be a greedy and unreliable approach, especially when dealing with a highly diverse viral community with many unknown viruses.

VirHostMatcher did not perform better when assigning the most frequent taxon among a more significant number of possible hosts (up to 30) with a score ≤ 0.25, contrary to what has been reported [14]. Instead, using this consensus criterion among the top 30 scoring hosts yielded a precision even lower than that of CrisprOpenDB and WIsH, which is known to perform better with incomplete contigs [16], while assigning host among the top 5 reached the second highest precision overall. Such discrepancies may depend on the distribution of taxa within the studied dataset. For instance, while increasing the n possible hosts criterion, one can expect a higher probability of finding multiple high-scoring instances of a particular host only by chance on a highly diverse dataset.

PHP allows predictions to be made with custom databases and provides a database of 60,105 bacterial genomes within the program's repository. Using this reference database, PHP obtains the second-lowest precision overall, while the custom database (133 bacterial genomes) elevated PHP as the most accurate and sensitive sequence composition method. This result implies that using an extensive reference database does not necessarily enhance the performance of virus-host prediction tools, unless the actual hosts are present. Thus, PHP may be a suitable tool, especially when working with metagenome-assembled genomes (MAGs) and metagenomic viral contigs (mVCs) from the same metagenome. Also, although not directly tested, host-dependent alignment-free methods such as PHP were noticeably more effortless to set up and faster to execute than integrative methods and virus and host-dependent alignment-based methods.

RaFAH achieved the highest precision, sensitivity, and F1_score on the test data collection. However, only a couple of its predictions on the metagenomic dataset were consistent with those of CRISPR-based methods, PHP, or the environment from which the metagenomes were generated. The metagenomic data analyzed here came from samples taken within the Cuatro Ciénegas Basin (CCB) which, despite being a desert oasis with oligotrophic waters, is known for sheltering diverse groups of microorganisms, many of which are endemic and

related to marine microorganisms [1, 7]. Such diversity is believed to have evolved as a result of the long-standing environmental stability of a deep aquifer that recreates an ancient ocean conditions, and which nourishes the aquatic systems of CCB through the movement of groundwater produced by the magmatic pouch deep in the Sierra San Marcos y Pinos [5]. Specifically, the environment from which samples were extracted is a shallow pond characterized by high pH and salinity known as Archaen Domes (AD) [8–10]. It has been shown that AD harbors a great diversity of bacteria on a short spatial scale [9] and is one of the most diverse archaeal communities in the world [8]. Such diversity includes sulfate-reducing *Proteobacteria* and extreme halophilic *Euryarchaeota* [42]. In addition, a highly diverse viral community has recently been described where haloarchaeaviruses constitute an essential part [10]. Therefore, predictions pointing to halophilic archaea, as well as halophilic, halotolerant, alkaliphilic, thermophilic, oligotrophic, sulfate-reducing, sulfur-oxidizing or marine bacteria, were considered consistent with the environment in question (Table 2).

Although using only spacers predicted from MAGs can result in dramatically lower sensitivity than relying on an extensive CRISPR spacers database, CrisprCustomDB produced more reliable predictions than CrisprOpenDB and was able to discriminate between possible hosts for contig C30N1L64 (further supported by PHP). On the one hand, this demonstrates the benefit of predicting hosts from *ad hoc* databases built using archaeal and bacterial MAGs from the same dataset, especially for highly diverse datasets that are likely to have a high proportion of novel viruses, such as the one tested here [10]. On the other hand, the fact that the ordinary CRISPR approach made more predictions on the metagenomic dataset than CrisprCustomDB probably reflects the benefit of using a more extensive spacer database (see Materials and Methods), as previously discussed regarding the performance of CrisprOpenDB. However, the lack of consistency of CrisprOpenDB and RaFAH with the other methods suggests that relying on a database of >11 million spacers [15] or a random forest classifier based on the protein content of viruses with a known host [18] may only be advantageous when the hosts or assembled viruses are already known or closely related to hosts or viruses represented in the respective databases. Therefore, we predict that incorporating spacers from MAGs into an extensive pre-compiled spacer database (which can be done with CrisprCustomDB) is likely to benefit from both approaches, increasing the precision and sensitivity of these predictions. It should be noted, however, that different selection criteria for MAGs (levels of contamination and completeness) and different spacer prediction tools may produce different results. For example, keeping only high quality MAGs may reduce false positives in spacer prediction but lose a lot of sensitivity, while using less stringent criteria for MAGs filtering may have the opposite effect, i.e. high sensitivity but a higher probability of false positives. Either way, this can result in spacer databases of different sizes, which we have seen can significantly affect host prediction.

The fact that PHP made the most predictions in agreement with other methods and the environment is consistent with the observation that the performance of alignment-free methods suffers less than that of alignment-based and virus-dependent methods when predicting the host of novel viruses [38]. Nevertheless, the CRISPR spacer-based methods collectively made three reliable predictions that PHP did not. RaFAH, on the other hand, made seven reliable predictions that PHP did not. This shows that although these methods are fundamentally different, they can complement and support each other. Incorporating these tools in an integrative software such as iPHoP [38], allows tackling the host prediction problem from different angles, increasing the chance of making the correct predictions. Also, judging the predictions based on the consistency between the predicted host biology (*i.e.*, taxonomy, habitat, lifestyle, or metabolism) and the source environment of the query virus (Table 2) may provide additional validation, mainly when predicting hosts of novel viruses. However, some caution still

needs to be exercised with this validation approach. For example, for predictions with less consistency between methods and at higher taxonomic ranks, there is an increased risk that the consistency between the source environment of mVCs and the biology of the predicted hosts will be rather ambiguous or even false.

Host prediction is one of the most critical features for characterizing mVCs, probably along with phylogenetic relationships. We wanted to know who the host is to learn more about the biology of the newly assembled virus, such as where it gets the resources to complete its replication cycle, what organisms it interacts with, and with whom it might co-evolve. However, although the host predictions presented here take us a step forward in characterizing AD viruses, we still need to know the phylogenetic context, the evolutionary processes, and the functional adaptations that will allow us to better understand the origin of diversity at this particular site.

## Conclusions

The results presented here indicated that RaFAH, a virus-dependent alignment-based method, and PHP, a host-dependent alignment-free method, are the best-performing tools for virus-host prediction. Other methods showed different performances depending on the host selection criteria, scoring thresholds, and the reference database. It seems that CRISPR-based methods seem to benefit from using a more extensive spacers database when predicting hosts of already-known viruses. However, using a more extensive candidate host database did not enhance the performance of host-dependent alignment-free methods such as PHP.

The synergy observed between PHP and CRISPR-based tools when applied to metagenomic viral contigs (mVCs) and metagenome-assembled genomes derived from the same dataset implies that the use of this combined approach, together with highly accurate methods that rely on extensive pre-compiled reference databases such as RaFAH, has the potential to produce more robust host assignments for exceptionally diverse metagenomic datasets. This assumes that predictions remain consistent across methods and that the taxonomy, habitat, lifestyle, or metabolism of the inferred host matches the characteristics of the source environment.

Finally, host predictions on mVCs from Archaean Domes showed that viruses inhabiting such environment infect halophilic archaea as well as a variety of bacteria which may be halophilic, halotolerant, alkaliphilic, thermophilic, oligotrophic, sulfate-reducing or marine-related. These predictions are consistent with the particular environment and the known geological and biological evolution of the Cuatro Ciénegas Basin and its microorganisms.

## Supporting information

**S1 File. Lists of NCBI complete bacterial virus genomes and RefSeq complete bacterial genomes used for benchmarking virus-host prediction tools.** The file contains six sheets—the first one lists bacteriophage genomes. The second is the Virus-Host DB table. The third one is the RefSeq release catalog with complete bacterial genomes—the fourth lists the virus-host pairs, including their accessions, used for benchmarking. The fifth sheet is the reference accession-genus list against which each prediction was compared. The sixth and final sheet contains the prediction results of each tool, using the parameters on which they perform the best.
(XLSX)

**S2 File. NCBI complete bacterial virus genomes and RefSeq complete bacterial genomes sequence files used for benchmarking virus-host prediction tools.** It includes viral and

bacterial genomes in fasta format. It also contains files with predicted CRISPR arrays and spacers.
(XLSX)

**S3 File. Files used for testing virus-host prediction tools on metagenomic data from Archaean Domes, Cuatro Ciénegas Basin, Mexico.** It includes metagenomic viral contigs and metagenome-assembled genomes in fasta format, as well as spacers needed to run predictions with BLAST and CrisprCustomDB and the host k-mer file needed to run predictions with PHP.
(XLSX)

**S4 File. Host prediction results on mVCs from Archaean Domes, Cuatro Ciénegas Basin, Mexico.** It includes a table with contig information and host predictions. Contigs highlighted with beige backgrounds will likely represent the same virus according to their protein domain content and host prediction. A red line delimits unreliable predictions from predictions supported by only one method but with consistency between the predicted host biology and the virus source environment. Above the yellow line, predictions are supported by two methods. Above the green line, predictions are supported by three methods.
(XLSX)

**S5 File. Host prediction results for complete NCBI archaea virus genomes.** It contains two tables. The first table shows the genus-level host predictions made by each tool for each archaea virus. False positive predictions are highlighted in red, while true positive predictions are highlighted in green. The second table shows the prediction statistics including precision, sensitivity and F1_score.
(XLSX)

## Acknowledgments

We greatly acknowledge Dr. Erika Aguirre-Planter, Dr. Rosalinda Tapia, Dr. Laura Espinosa-Asuar, Jazmín Sánchez-Pérez, Mariette Viladomat-Jasso, Rodrigo Vázquez, and Manuel Rosas from the Instituto de Ecología, UNAM, for their logistical and technical support and assistance during sample collection, and laboratory processing. We also acknowledge Dr. José Alberto Campillo-Balderas for reviewing the manuscript, and Rodrigo García-Herrera for providing access to the LANCIS-Instituto de Ecología's Supercomputer Unit at UNAM. Finally, we thank SEMARNAT and APFF Cuatro Ciénegas for facilitating the sampling and, in particular, Rancho Pozas Azules, PRONATURA Noreste for access and permission to sample in the Cuatro Ciénegas Basin Natural Protected Area.

## Author Contributions

**Conceptualization:** Alejandro Miguel Cisneros-Martínez, Ulises E. Rodriguez-Cruz, Luis E. Eguiarte, Valeria Souza.

**Data curation:** Alejandro Miguel Cisneros-Martínez.

**Formal analysis:** Alejandro Miguel Cisneros-Martínez.

**Funding acquisition:** Luis E. Eguiarte, Valeria Souza.

**Methodology:** Alejandro Miguel Cisneros-Martínez, Ulises E. Rodriguez-Cruz, Luis E. Eguiarte, Valeria Souza.

**Software:** Alejandro Miguel Cisneros-Martínez.

**Supervision:** Luis D. Alcaraz, Arturo Becerra, Luis E. Eguiarte, Valeria Souza.

**Visualization:** Alejandro Miguel Cisneros-Martínez.

**Writing – original draft:** Alejandro Miguel Cisneros-Martínez.

**Writing – review & editing:** Luis D. Alcaraz, Arturo Becerra, Luis E. Eguiarte, Valeria Souza.

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
