## [Decision Letter · Decision Letter 0]

31 Oct 2023

PONE-D-23-27755Comparative evaluation of bioinformatic tools for virus-host prediction and their application to a highly diverse community in the Cuatro Ciénegas Basin, MexicoPLOS ONE

Dear Dr. Souza,

Thank you for submitting your manuscript to PLOS ONE. After careful consideration, we feel that it has merit but does not fully meet PLOS ONE’s publication criteria as it currently stands. Therefore, we invite you to submit a revised version of the manuscript that addresses the points raised during the review process.

Please see below the comments and suggested MAJOR revisions made by the individual(s) who reviewed your manuscript.  If provided, the referee's report(s) indicate the revisions that need to be made before it can be accepted for publication.

We look forward to receiving your revised manuscript.

Kind regards,

Ricardo Santos

Academic Editor

PLOS ONE

Journal Requirements:

3. Please expand the acronyms “CONAHCYT, DGAPA/UNAM-PAPIIT, and ANID-CEQUA” (as indicated in your financial disclosure) so that it states the name of your funders in full.

"I have read the journal's policy and the authors of this manuscript have the following competing interests: LDA is an Academic Editor for this journal. This does not alter our adherence to the PLOS ONE policy on sharing data and materials."

6. Please include a new copy of Tables 1 and 2 in your manuscript; the current table is difficult to read. Please follow the link for more information: https://blogs.plos.org/plos/2019/06/looking-good-tips-for-creating-your-plos-figures-graphics/

7. We are unable to open your Supporting Information file [S3 File]. Please kindly revise as necessary and re-upload.

8. Please upload a copy of Supporting Information Figure/Table/etc. S2 File which you refer to in your text on page 9.

Reviewers' comments:

Reviewer's Responses to Questions

**Comments to the Author**

1. Is the manuscript technically sound, and do the data support the conclusions?

Reviewer #1: Yes

Reviewer #2: Partly

2. Has the statistical analysis been performed appropriately and rigorously? 

Reviewer #1: Yes

Reviewer #2: Yes

3. Have the authors made all data underlying the findings in their manuscript fully available?

Reviewer #1: Yes

Reviewer #2: No

4. Is the manuscript presented in an intelligible fashion and written in standard English?

Reviewer #1: Yes

Reviewer #2: Yes

5. Review Comments to the Author

Reviewer #1: The manuscript titled "Comparative evaluation of bioinformatic tools for virus-host prediction and their application to a highly diverse community in the Cuatro Ciénegas Basin, Mexico" described the comparing virus-host prediction tools including authors' challenge. The study provides valuable insights into the performance of various tools and methodologies in the field.

Miscellaneous comments:

1. Authors need to add the full name of TARA in the manuscript.

2. Please consider to move Fig 1 into the result section.

Reviewer #2: General review:

The authors present a comparison between different host-virus prediction tools, as well as their own tool, CrisprCustomDB, “inspired” by another tool, CrisprOpenDB. Firstly, it is not explained how CrisprCustomDB works precisely, besides the criteria used for host selection, which are suggested as being the same as in CrisprOpenDB. More details should be presented as to how the tool functions and how it differentiates from CrisprOpenDB. Moreover, the results from the implemented tool could be considered underwhelming and show no significant differences with other tools or strategies that the authors cite. Therefore, a more concrete explanation of what advantages CrisprCustomDB presents is needed.

Particular comments:

Fig 1: The "integrative methods" category should be added to the figure.

Line 136: the authors state that “we believe that the fact that host-dependent methods do not necessarily depend on a sizeable pre-compiled reference database represents an advantage”. It is not clear enough what is the advantage the authors are referring to: if it is related to having smaller databases or being able to compile a custom database, to include MAGs. In the first case, using only MAGs could result in lower sensitivity, since there are always genomes in a metagenome that cannot be reconstructed or are bad quality genomes, and could result in lower precision and sensitivity, depending on the quality of the MAGs reconstructed. On the other hand, if the advantage is to have customizable databases that allow the inclusion of MAGs, these can be more "sizeable" than some pre-compiled reference databases.

Line 165: The S2_File.zip in GitHub does not contain any of the information cited in the manuscript. How were the 133 genomes selected? Table S1 shows that for different species the number of genomes used ranges between 1 and 9. Given that there are more genomes available for most of the species used, more information about how this selection was made is necessary to better understand the difference between the number of genomes used and the number of genomes with spacers.

Line 170: The explanation about false positives, false negatives and the two null hypothesis is confusing and maybe not that necessary. I recommend rewriting it to make it more concise and clear.

Line 189: It is not exactly clear how the script implemented works and what is its novelty. It says that it was "inspired by CrisprOpenDB" and uses the same criteria to determine the virus' host. The main advantage of the presented script seems to be the possibility to use a custom database, but it is shown that custom databases underperform when compared to CrisprOpenDB. Using predicted CRISPR spacers from an environment to expand existing databases could be an interesting strategy to improve a prediction, as is mentioned in line 463 and 500. However, limiting the database to only predicted CRISPR spacers from MAGs risks having poor results, given the difficulties to assemble CRISPR systems from metagenomics and the wide diversity in quality that MAGs tend to show. If only high quality MAGs are used, the number of sequences in the database decreases and the sensitivity of the tool can fall; instead, if draft MAGs are included, there is risk to increase the number of false positives, due to misbinned contigs.

Line 191: "host if spacers have a maximum of 2 mismatches with the viral genome". The start of the phrase should be more clear.

Line 195: in line 192 the authors use the phrase "criteria 1", in line 194 "criterion 2" and in this line "criteria three". I recommend always using the same approach, for example "criterion 1", "criterion 2", etc.

Line 255: All MAGs were considered or only those with determined completeness and contamination values? It should be discussed also how different criteria for MAG selection could affect results.

Line 267: The double hyphen (--) is shown as a long hyphen

Line 270: Why two different tools were used to obtain spacers (CRISPRCasTyper and CRISPRDetect)? Do they provide different results?

Table 1 shows a "NA" column, but it is not clear what it means. The values in the NA column are calculated as the difference between predicted and actual virus-host pairs, but this difference is influenced by the number of false positives and there is already a "False negative" column.

Fig 2: Could including the type of method (host/virus dependent, alignment based/free) to the figure make it easier to see the comparisons mentioned later (line 311)?

Line 382: Some RaFAH results were not supported by both other methods and the environmental information that the authors mention in the discussion. Given that RaFAH showed the best results with the benchmarking dataset, it could be interesting to mention in the results section the environmental analysis carried out, and maybe adding a figure, to illustrate the short-comings of RaFAH when applied to a diverse environment like the one used in this work.

Line 395: It could be informative to clarify what were the computational limitations: disk space, memory required, processing time, etc.

Line 413: There is no closing parenthesis

Line 420: Given that in the introduction Archaea are mentioned as important for the studied environment, it could be useful to make this clarification in the Materials and Method section where the benchmarking dataset is described. However, given the importance given to Archaea and the lack of results described by the other tools authors, it could be an interesting supplementary result, even if few data is available.

Line 503: Even though the combination of tools show improved results, CrisprCustomDB only added 2 new hits to the 54 results of PHP, and those hits were also identified using BLAST against the predicted spacers. It is still difficult to see the advantage of using CrisprCustomDB instead of other tools.

Line 531: This paragraph is one extremely long sentence. Authors should consider re-writing it.

File_S1: There are some lines in the file that were not separated in columns.

File_S4: what does the "Coverage" column refer to?

6. PLOS authors have the option to publish the peer review history of their article (what does this mean?). If published, this will include your full peer review and any attached files.

Reviewer #1: No

Reviewer #2: No

---

## [Author Response · Author response to Decision Letter 0]

4 Dec 2023

-We have red PLOS ONE's style requirements and we believe that our manuscript and files meet the formatting requirements.

-We have deposited our data in a Figshare repository (https://doi.org/10.6084/m9.figshare.24649935.v2). Deposited data include S2_File and S3_File.

3. Please expand the acronyms “CONAHCYT, DGAPA/UNAM-PAPIIT, and ANID-CEQUA” (as indicated in your financial disclosure) so that it states the name of your funders in full.

-We have expanded acronyms to include the name of our funders in full. Modified financial disclosure: AMCM is a Ph.D. student in the Programa de Doctorado en Ciencias Biomédicas, Universidad Nacional Autónoma de México (UNAM) and has received the Consejo Nacional de Humanidades, Ciencias y Tecnologías (CONAHCYT) grant 814975. This study was supported by Programa de Apoyo a Proyectos de Investigación e Innovación Tecnológica (PAPIIT) – Dirección General de Asuntos del Personal Académico (DGAPA) - UNAM (grant numbers IG200319 and IN204822). This manuscript was written with Chile Agencia Nacional de Investigación y Desarrollo (ANID) – Centro de Estudios del Cuaternario de Fuego-Patagonia y Antártica (CEQUA) R-20F0009 to VS and LEE as part of a larger comparison with ocean waters in the southern tip of the American continent. The funders had no role in study design, data collection and analysis, decision to publish, or preparation of the manuscript.

"I have read the journal's policy and the authors of this manuscript have the following competing interests: LDA is an Academic Editor for this journal. This does not alter our adherence to the PLOS ONE policy on sharing data and materials."

-Our Competing Interests statement already included the confirmation that competing interests do not alter our adherence to all PLOS ONE polices on sharing data and materials: “I have read the journal's policy and the authors of this manuscript have the following competing interests: LDA is an Academic Editor for this journal. This does not alter our adherence to the PLOS ONE policy on sharing data and materials”.

-We have modified our Data Availability statement to include the specifications on where to find the study's minimal dataset: Supporting data are provided as supplementary data files. S1_File, S4_File, and S5_File are available with the online version of this article. The minimal data set required to replicate the reported study findings in their entirety, constituted by S2_File and S3_File, is available via the public Figshare repository: https://doi.org/10.6084/m9.figshare.24649935.v2. Metagenomic viral contigs and metagenome-assembled genomes have been submitted to GenBank under BioProject accession PRJNA847603 and will be released to the public database once processed by GenBank annotation staff.

6. Please include a new copy of Tables 1 and 2 in your manuscript; the current table is difficult to read. Please follow the link for more information: https://blogs.plos.org/plos/2019/06/looking-good-tips-for-creating-your-plos-figures-graphics/

-We have included a new copy of Tables 1 and 2 in our manuscript so that they fit the document margins.

7. We are unable to open your Supporting Information file [S3 File]. Please kindly revise as necessary and re-upload.

-We have revised S3_File and uploaded it to the Figshare repository https://doi.org/10.6084/m9.figshare.24649935.v2.

8. Please upload a copy of Supporting Information Figure/Table/etc. S2 File which you refer to in your text on page 9.

-S2_File was originally uploaded to a Github repository because it exceeded the size limits of the editorial manager portal. Now it can be found in the Figshare repository https://doi.org/10.6084/m9.figshare.24649935.v2.

Reviewers comments:

Reviewer #1: The manuscript titled "Comparative evaluation of bioinformatic tools for virus-host prediction and their application to a highly diverse community in the Cuatro Ciénegas Basin, Mexico" described the comparing virus-host prediction tools including authors' challenge. The study provides valuable insights into the performance of various tools and methodologies in the field.

Miscellaneous comments:

1. Authors need to add the full name of TARA in the manuscript.

-Line 84-86. We added the full name of TARa several expeditions.

2. Please consider to move Fig 1 into the result section.

-Line 276. Fig 1 was moved into the result section.

Reviewer #2: General review: The authors present a comparison between different host-virus prediction tools, as well as their own tool, CrisprCustomDB, “inspired” by another tool, CrisprOpenDB. Firstly, it is not explained how CrisprCustomDB works precisely, besides the criteria used for host selection, which are suggested as being the same as in CrisprOpenDB. More details should be presented as to how the tool functions and how it differentiates from CrisprOpenDB. Moreover, the results from the implemented tool could be considered underwhelming and show no significant differences with other tools or strategies that the authors cite. Therefore, a more concrete explanation of what advantages CrisprCustomDB presents is needed.

-Line 211-247. A justification for developing CrisprCustomDB, as well as a more detailed explanation of its functioning is presented in the Materials and Methods section.

-Line 567-573. A more concrete explanation of the advantages of using CrisprCustomDB is presented. 

Particular comments:

Fig 1: The "integrative methods" category should be added to the figure.

-Fig 1 has been modified to include “integrative methods” category.

Line 136: the authors state that “we believe that the fact that host-dependent methods do not necessarily depend on a sizeable pre-compiled reference database represents an advantage”. It is not clear enough what is the advantage the authors are referring to: if it is related to having smaller databases or being able to compile a custom database, to include MAGs. In the first case, using only MAGs could result in lower sensitivity, since there are always genomes in a metagenome that cannot be reconstructed or are bad quality genomes, and could result in lower precision and sensitivity, depending on the quality of the MAGs reconstructed. On the other hand, if the advantage is to have customizable databases that allow the inclusion of MAGs, these can be more "sizeable" than some pre-compiled reference databases.

-Such statement was removed from the introduction. Relevant discussion on the subject was added on line 581-591.

Line 165: The S2_File.zip in GitHub does not contain any of the information cited in the manuscript. How were the 133 genomes selected? Table S1 shows that for different species the number of genomes used ranges between 1 and 9. Given that there are more genomes available for most of the species used, more information about how this selection was made is necessary to better understand the difference between the number of genomes used and the number of genomes with spacers.

-Line 118-128. Materials and Methods section was modified to clarify data selection.

Line 170: The explanation about false positives, false negatives and the two null hypothesis is confusing and maybe not that necessary. I recommend rewriting it to make it more concise and clear.

-Line 129-136. The explanation about false positives and false negatives was rewritten to make it more concise and clear.

Line 189: It is not exactly clear how the script implemented works and what is its novelty. It says that it was "inspired by CrisprOpenDB" and uses the same criteria to determine the virus' host. The main advantage of the presented script seems to be the possibility to use a custom database, but it is shown that custom databases underperform when compared to CrisprOpenDB. Using predicted CRISPR spacers from an environment to expand existing databases could be an interesting strategy to improve a prediction, as is mentioned in line 463 and 500. However, limiting the database to only predicted CRISPR spacers from MAGs risks having poor results, given the difficulties to assemble CRISPR systems from metagenomics and the wide diversity in quality that MAGs tend to show. If only high quality MAGs are used, the number of sequences in the database decreases and the sensitivity of the tool can fall; instead, if draft MAGs are included, there is risk to increase the number of false positives, due to misbinned contigs.

-Line 211-247. A justification for developing CrisprCustomDB, as well as a more detailed explanation of its functioning is presented in the Materials and Methods section.

-Line 567-573. A more concrete explanation of the advantages of using CrisprCustomDB is presented.

-Table 1 and Fig 2. CrisprCustomDB was removed due to its underwhelming performance. We are no longer presenting CrisprCustomDB as an alternative to existing virus-host prediction tools, but as a custom script developed to answer hypothesis specified in line 219.

-Line 513-518. Data associated to Table 1 and Fig 2, regarding CrisprCustomDB, is now discussed in the Discussion section.

Line 191: "host if spacers have a maximum of 2 mismatches with the viral genome". The start of the phrase should be more clear.

-Line 237-247. A detailed explanation of CrisprCustmDB host selection criteria is included.

Line 195: in line 192 the authors use the phrase "criteria 1", in line 194 "criterion 2" and in this line "criteria three". I recommend always using the same approach, for example "criterion 1", "criterion 2", etc.

-Line 237-247. Criterion 1, criterion 2, and criterion 3 are explained.

Line 255: All MAGs were considered or only those with determined completeness and contamination values? It should be discussed also how different criteria for MAG selection could affect results.

-Line 200-203. Now we mention the total number of MAGs and the number of MAGs with high quality.

-Line 229-230. Here we mention the number of MAGs used for CRISPR spacer prediction.

-Line 584-591. Here we discuss how different criteria for MAG selection could affect results.

Line 267: The double hyphen (--) is shown as a long hyphen

-Line 251. Long hyphen was replaced by double hyphen.

Line 270: Why two different tools were used to obtain spacers (CRISPRCasTyper and CRISPRDetect)? Do they provide different results?

-Line 226-229. Here we explain why we used CRISPRDetect.

-Line 248-250. Here we explain why we implemented a second CRISPR spacer-based approach.

Table 1 shows a "NA" column, but it is not clear what it means. The values in the NA column are calculated as the difference between predicted and actual virus-host pairs, but this difference is influenced by the number of false positives and there is already a "False negative" column.

-Line 134-135. The meaning of “NA” is specified.

Fig 2: Could including the type of method (host/virus dependent, alignment based/free) to the figure make it easier to see the comparisons mentioned later (line 311)?

-Fig 2 was modified to include the type of method.

Line 382: Some RaFAH results were not supported by both other methods and the environmental information that the authors mention in the discussion. Given that RaFAH showed the best results with the benchmarking dataset, it could be interesting to mention in the results section the environmental analysis carried out, and maybe adding a figure, to illustrate the short-comings of RaFAH when applied to a diverse environment like the one used in this work.

-Line 370-459. The results section was modified to include the environmental analysis carried out.

-S4_File was modified to include only a table. The associated figure in S4_File (a Venn diagram) was removed.

-Fig 3 was created.

Line 395: It could be informative to clarify what were the computational limitations: disk space, memory required, processing time, etc.

-Line 479. Computational limitations are clarified.

Line 413: There is no closing parenthesis

-Line 497-499. Long hyphens were added to make the parentheses-framed sentence clearer.

Line 420: Given that in the introduction Archaea are mentioned as important for the studied environment, it could be useful to make this clarification in the Materials and Method section where the benchmarking dataset is described. However, given the importance given to Archaea and the lack of results described by the other tools authors, it could be an interesting supplementary result, even if few data is available.

-Line 126-127. The number of genomes used for the archaea benchmarking dataset is mentioned in the Materials and Methods section.

-Line 503-506. The results on the archaea benchmarking dataset are briefly described.

-S5_File was created.

Line 503: Even though the combination of tools show improved results, CrisprCustomDB only added 2 new hits to the 54 results of PHP, and those hits were also identified using BLAST against the predicted spacers. It is still difficult to see the advantage of using CrisprCustomDB instead of other tools.

-Line 439-442. The benefit of using CrisprCustomDB over the standard CRISPR approach is mentioned.

-Line 567-584. The benefit of including data derived from MAGs is discussed. 

-Line 592-600. The emphasis was changed to highlight, not the benefit of using CrisprCustomDB, but the benefit of following a complementary approach.

Line 531: This paragraph is one extremely long sentence. Authors should consider re-writing it.

-Line 625-632. Paragraph was rewritten.

File_S1: There are some lines in the file that were not separated in columns.

-We believe that the reviewer may be referring to text separated by commas as in a .csv file. However, in this case comma-delimited text is intentionally part of the same column. For example, a list of accessions related to the same bacteria.

File_S4: what does the "Coverage" column refer to?

-It is a metric derived from the viralComplete script (part of metaviralSPAdes) when assessing the assembled viral genome integrity. However, as we deem it irrelevant in the present context the whole column was removed.

---

## [Decision Letter · Decision Letter 1]

28 Dec 2023

Comparative evaluation of bioinformatic tools for virus-host prediction and their application to a highly diverse community in the Cuatro Ciénegas Basin, Mexico

PONE-D-23-27755R1

Dear Dr. Souza,

We’re pleased to inform you that your manuscript has been judged scientifically suitable for publication and will be formally accepted for publication once it meets all outstanding technical requirements.

Kind regards,

Ricardo Santos

Academic Editor

PLOS ONE

Additional Editor Comments (optional):

Reviewers' comments:

Reviewer's Responses to Questions

**Comments to the Author**

1. If the authors have adequately addressed your comments raised in a previous round of review and you feel that this manuscript is now acceptable for publication, you may indicate that here to bypass the “Comments to the Author” section, enter your conflict of interest statement in the “Confidential to Editor” section, and submit your "Accept" recommendation.

Reviewer #1: All comments have been addressed

Reviewer #2: All comments have been addressed

2. Is the manuscript technically sound, and do the data support the conclusions?

Reviewer #1: Yes

Reviewer #2: Yes

3. Has the statistical analysis been performed appropriately and rigorously? 

Reviewer #1: Yes

Reviewer #2: Yes

4. Have the authors made all data underlying the findings in their manuscript fully available?

Reviewer #1: Yes

Reviewer #2: Yes

5. Is the manuscript presented in an intelligible fashion and written in standard English?

Reviewer #1: Yes

Reviewer #2: Yes

6. Review Comments to the Author

Reviewer #1: Thank you to revise the manuscript. Please provide high quality figures. Current version of figures are difficult to read.

Reviewer #2: 517: CrisprOpenDB is misspelled

File S1, tab "bacterialViruses": the following lines show a different format than the others (maybe they were not separated in columns): 326, 327, 353, 354, 609, 610, 611, 758, 759, 953, 954, 1741, 1742, 2911, 2912, 2913, 2914, 3101, 3102, 3103, 3105, 3106, 3107, 3111, 3112, 3113, 3117, 3118, 3119, 3121, 3122, 3123, 3132, 3133, 3134, 3143, 3144, 3145, 3956, 3957, 3958, 3961, 3962, 3964, 3965, 3967, 3968, 3969, 3970, 3975, 3976, 3983, 3984, 3993, 3994, 3996, 3997

7. PLOS authors have the option to publish the peer review history of their article (what does this mean?). If published, this will include your full peer review and any attached files.

Reviewer #1: No

Reviewer #2: No

---

## [Editor Report · Acceptance letter]

24 Jan 2024

PONE-D-23-27755R1 

PLOS ONE

Dear Dr. Souza, 

I'm pleased to inform you that your manuscript has been deemed suitable for publication in PLOS ONE. Congratulations! Your manuscript is now being handed over to our production team.

Kind regards, 

on behalf of

Dr. Ricardo Santos 

Academic Editor

PLOS ONE